# Priority Setting in the Polish Health Care System According to Patients’ Perspective

**DOI:** 10.3390/ijerph18031178

**Published:** 2021-01-28

**Authors:** Anna Rybarczyk-Szwajkowska, Izabela Rydlewska-Liszkowska

**Affiliations:** Department of Management and Logistics in Health Care, Faculty of Health Sciences, Medical University of Lodz, 90-131 Lodz, Poland; izabela.rydlewska-liszkowska@umed.lodz.pl

**Keywords:** setting priorities, health priorities, patients’ opinion, health care financing, health care system

## Abstract

Identification of health priorities is concerned with equitable distribution of resources and is an important part of strategic planning in the health care system. The aim of this article is to describe health priorities in the Polish health care system from the patients’ perspective. The study included 533 patients hospitalized in the Lodz region. The average age of the respondents was 48.5 years and one third (36.6%) had university education. Most of the respondents (64.9%) negatively assessed the functioning of the health care system in Poland. Most of them claimed the following aspects require improvements: financing health services (85.8%), determining priorities in health care (80.3%), the role of health insurance (80.3%), and medical education (70.8%). Over 70% of the respondents agreed the role of politicians in designing and implementing health system reforms should be limited. The fact that the respondents so negatively assessed the Polish health care system implies there is a need for full discussion on redefining health priorities.

## 1. Introduction

Reforming the health care system involves social, political, and economic considerations. It requires a robust process to efficiently and effectively establish priorities to guide the reform, and to ensure accountability and equity [1,2]. Establishing priorities is one of the greatest challenges faced by policy makers [3,4]. A priority is an instruction to help define the most critical activities, and signals the important work of an organization or system [5]. Prioritizing is essential when demands and needs exceed resources. Priority setting decisions are made at different levels [4,5,6,7], including national, regional, system, and institution [8], and often lack appropriate tools, information, and processes. The use of scientific evidence and principles in setting health priorities has enormous potential to lead to more rational decision making, especially in low resource settings [9,10,11]. Health policy in Poland has changed and the quality of services has increased, as well as the level of financing, mainly from public benefits. Despite a constant growth of indexes reflecting the health status of Polish society, just as in the majority of European countries, Polish society is growing older. This implies a necessity to reorganize the system. Polish health care is far from being ideal. In the rankings, in comparison with other countries, as well as the neighboring ones, its picture is bad. Patients, physicians themselves, and politicians complain about it [12]. The present situation of the Polish health care system has been and is still influenced by numerous political, market, economic, and human factors. The changes that have been introduced still do not seem to fully include the superior role of the patient in the system. Presently, economic and demographic factors seem to mostly affect the health care system. [13]. Polish health care priorities are established mainly to ensure this equitable distribution of resources, which is an important part of strategic planning [14]. However, a properly functioning health care system must also prioritize aspects of health care delivery valued by patients. With social solidarity and equity being overarching priorities of the nation, the health care system must define its own priorities, with larger national goals, that will deliver quality care and meet expectations of consumers [13]. Because defined priorities drive organizational performance, they must be carefully selected. The question then arises as to who should be involved in this process of defining priorities and what method will produce the most effective and ethical results. Involving all the stakeholders in the system, including the public and local community, in selecting priorities, especially in publicly funded programs and initiatives [15], is an ideal solution. Contributions made by all the stakeholders of the system could have a significant impact on the quality and efficiency of the health care system. 

The aim of the article is to present the views of the health care system beneficiaries on the priorities that should be adopted by the Polish health care system. 

## 2. Materials and Methods 

The survey results presented in the article are one of the stages of an international research program on conditions ensuring effectiveness and efficiency of health care systems in Poland and Ukraine, including the main problems and research areas, which is aimed at promoting optimal solutions for organization, functioning, and financing of the health care system. 

The pilot study on 22 people was carried out in May 2017. Next, a face-to-face interview questionnaire was conducted on 533 respondents (18 years of age and older), who agreed to participate in this research, between September 2017 and March 2018, in four public health care units in the Lodz region. There were several reasons to conduct the study in a selected group of patients. The study was the research stage aimed at improving the research tool and involved obtaining the highest possible number of answers within the existing financial constraints before further research on a randomly selected sample (the research was not financed from external funds). The aim of the study was also to learn about patients’ opinions that were dominant in the structure of responses in order to possibly strengthen the role of some of the questions asked in the future questionnaire. A goal of the study was to include as many participants as possible up to 500. At the initial stage, no formal calculation of sample size was carried out but we chose a group of more than 100 patients in each of the hospitals. A similar number of patients was in every group. The patients were selected according to the acceptance criteria: they did not feel pain, they felt relatively well, and agreed to participate in our study. The participation was voluntary and anonymous. All the subjects had given their informed consent before inclusion in the study. The interviews took from 20 to 30 min. 

The interviewers were granted written consent by directors and heads of wards to conduct the research in hospitals. 

The research tool was a questionnaire (Appendix A) prepared by the staff of the Department of Health Care Policy and the Centre for Research on Health Care Strategies and Health Policy at the Warsaw School of Economics in cooperation with the National Medical University in Kiev. 

The questionnaire consisted of twenty-four questions: demographic characteristics of the respondents, the respondents’ self-rated standards of living and health assessment, interest in health care system changes, frequency of receiving health care and the form of health care the respondents are most frequently provided with. Questions about patients’ opinion of optimal solutions for the organization, functioning, and financing of the health care system in Poland made up the second, more extensive part of the questionnaire.

Each of the listed categories was assessed on a 7-pont Likert scale from “definitely bad/definitely not” (-3 points), “no” (-2 points), “somewhat not” (-1 point), “no opinion” (0), “somewhat yes” (1 point), “yes” (2 points), “definitely good/definitely yes” (3 point). There were no open questions in the questionnaire. The questionnaire was evaluated in terms of compliance with the main rules for questionnaire construction, i.e., simple vocabulary, avoiding words that are abstract, not fully defined, or ambiguous, and avoiding jargon, moralizing language, negations, names of institutions, surnames, and lengthy items.

Data were analyzed using Statistica software (No. 13.1.336.0) and Microsoft Excel 2018. The level of statistical significance was set at *p* < 0.05. The frequency, mean, standard deviation, and percentages were calculated.

## 3. Results

### 3.1. Characteristics of the Respondents

The study included 533 patients, 330 (61.9%) females and 203 (38.1%) males. The mean age of the respondents was 48.5 years and individuals with a university (34%) or secondary (33%) education level prevailed in the group. Over half of the subjects were married (54%). Among the study participants, 17% had the status of disability or old-age pensioner, and almost half of them were employed under an employment contract or other contracts. Nearly two-thirds of the study participants (63%) regarded their financial status to be rather good or good. Another personal data question referred to their self-rated health status. Among the study participants, 68% considered their health condition quite good, good, or very good (35%, 27%, 6%, respectively). Only one fifth of the respondents assessed their health condition negatively. Then, the patients were asked whether they were interested in information on changes in the health care system. A great majority of the participants (72%) gave the answer “yes”, whereas 13% of the patients had no opinion on this issue. The last personal data questions referred to frequency of receiving health care and forms of care the patients were most often provided with. Characteristics of the study sample are shown in Table 1.

### 3.2. Health Priorities According to Patients’ Opinions

This paper focuses on responses to survey questions related to views on the health care system and health system priorities. The respondents were asked to provide a general assessment of the health care system in Poland. Most participants (64.9%) had negative views; only 19.6% of the respondents assessed the system positively. No respondent gave the answer *definitely good* or *does not require any changes*, while 15.5% of the respondents had no opinion on the issue.

The respondents were asked whether they agreed, and if so to what extent, with the selected aspects of the public health care system. The questionnaire included the following statements: (1) patients are treated with kindness and care, (2) there are no problems with making an appointment with a primary care doctor, (3) it is easy to obtain information on access to health benefits, (4) medical treatment is fully free, (5) treatment conditions are good, (6) doctors are willing to give referrals to medical specialists if the patient’s condition requires such, (7) patients can expect immediate medical assistance, (8) all patients are treated equally. It is difficult to unequivocally establish whether the respondents agree with all the statements due to divergent opinions. However, it needs to be emphasized that the opinions on financing of the health care system were the least divergent. Almost 60% of the respondents did not agree with the statement that *treatment is fully free*. 

The quality and accessibility of health benefits are extremely important aspects of the evaluation of the health care system. In the study, the respondents were asked about their opinions on the aforementioned aspects of the health care system with regard to both public and private institutions. On the basis of the collected data, it is not possible to unequivocally determine whether the opinions on the quality and accessibility of health services are positive or negative. Opinions on these aspects in the private sector are an exception—they were positively evaluated by a great majority of the respondents (quality 78.6% and accessibility 80%). 

The patients were also asked whether factors such as (1) organization of the health care system, (2) financing, (3) the number of practicing doctors, (4) competences of practicing doctors, (5) hospital infrastructure, (6) medical equipment in diagnostics and therapy, (7) costs of medications, and (8) prevention/health education affect the efficiency of the health care system. The obtained results allow us to draw a conclusion that the patients share the same opinions. Over 70% of the respondents believe that these factors do have an impact on the efficiency of the system, whereas the majority of the respondents agree with the statement that financial resources make up the most significant factor. 

The respondents were also asked to express their opinion on the factors/areas which require changes and influence the efficiency of the health care system. A great majority of them believe that the aspects requiring radical changes include: financing of health services (85.8%), priorities in the health care system (80.3%), the role of health insurance (80.3%), the system of training for medical personnel (70.8%), and free-market rules (64.5%). One of the assessed elements was the role of the state (the ruling party) in the decision-making process in the system. Over 70% of the respondents believe that the role of politicians in planning and implementing reforms in the health sector should be limited. Table 2 presents the results.

## 4. Discussion

The priority setting in health care remains a crucial part of health care planning and resource allocation. Besides, it is what patients pay attention to, observe, and assess from their own perspective [14,15,16]. The approach adopted in the article differs from the survey on patient satisfaction as it focuses on “everyday” beneficiary patients’ reflections on their experiences in using health services concerning care, organization of service delivery, financing, accessibility to services, and quality of services. Moreover, patients’ opinions on potential changes in the health care system in Poland reveal their attitudes towards the current state of the system. Although the obtained results are not surprising for some researchers and health care stakeholders, they are valuable evidence of patients’ opinions which may trigger the process of meeting public expectations and setting directions for further research. However, it is difficult to separate these areas dichotomously.

The obtained results of the survey of patients’ opinions may make it possible to identify effective and patient-friendly solutions in health care, but also allow the recognition of limitations while implementing health services [17]. 

Negative opinions on the Polish health care system, expressed by over half of the respondents, as well as the fact that a great majority of the respondents claim that the aspects presented in the study require immediate improvement, are clear proof of dissatisfaction with the way health care is delivered, particularly demonstrated in payments for health care services and unfair treatment of patients. Patients’ personal opinions on more detailed issues of the health care system and identifying key problems with them may facilitate interpretation of the overall opinion on the health care system. The gradual refinement of questions included in the questionnaire may be helpful in better understanding the respondents’ views and enable interpretation of the reasons for their answers [18]. 

The tasks of the state authorities aimed at providing equal access to health care in Poland are specified in the Law of 27 August 2004 on Health Care Benefits Financed from Public Funds [19]. These tasks include providing conditions for the functioning of the health care system; analysis and identification of health needs and factors changing them; health promotion and prevention focused on creating conditions favorable for health; and financing. It needs to be emphasized that financing is the very element of the health care system that affects its functioning [20,21]. The majority of individuals claim that financing is a factor which has the greatest influence on the effectiveness of the system and believe that the rules of the financing of health benefits require radical changes. Consequently, they share the opinion that the role of health insurance should be changed radically. Given the patients’ perception and results of research on health care accessibility [22], it could be stated that the amount of money spent on health care in Poland does not make health benefits more available nor does it satisfy the medical needs of Polish patients. Health care stakeholders debate whether the subject of resource allocation should be reinitiated. 

Patients also pay particular attention to medical equipment used in diagnostics and therapy, as well as prevention and education. They consider them important factors influencing the effectiveness of health care. While the first factor seems to be related to the accessibility of health technologies, which in turn may have various factors (financial, organizational, administrative), the second factor shows a quite different value for patients who want to improve their knowledge in the field of health protection. It is reflected in the active attitude of the respondents towards factors and areas requiring changes, mainly training medical personnel, financing, insurance, and priorities.

The opinions of members of society can help to identify health care areas for the action of policy makers, medical professionals, and other bodies responsible for setting health care priorities. Besides, they can contribute to establishing directions of redefining methods and instruments for increasing patient satisfaction. However, we should bear in mind the fundamental difference between patient satisfaction and meeting their expectations in terms of their health needs [23,24,25,26,27,28,29].

Moreover, the feedback obtained from patients may inspire the creation of methods of promoting and convincing patients of the nationally chosen priorities by a dialogue platform or disseminating knowledge on opportunities and limitations that the health care system may meet, assuming scarce system resources [30]. It seems to be indisputable that the assumptions of future health policy should be formulated in the context of results of the survey that reflects the opinions of different social groups, including beneficiaries of the health care system.

### Limitations

Our study had limitations. Firstly, we described priority setting as perceived by patients, which may not reflect the objective priority-setting process. However, the information obtained from our participants may be an indicator of the degree of publicity of the priority-setting process, which itself is a relevant finding for this study. Secondly, we are not able to generalize our findings but they provided us with insight in the priority setting of the cases we described. Thirdly, the results may be imprecise due to a small number of questions in the survey. This issue requires further research.

## 5. Conclusions

Identification of health priorities may improve the functioning and management of changes in the health care system, provided that they constitute a unified set of objectives.Beneficiaries of the health care system strongly claim that the role that politicians play in planning and implementing reforms in the health sector should be limited.One top priority constraint on the health care system is its financing—a factor that has the most significant impact on the efficiency and functionality of the system. Beneficiaries surveyed in this study share the opinion that the means of financing health benefits require immediate changes, which should reopen a discussion on the health insurance system.

## Figures and Tables

**Table 1 ijerph-18-01178-t001:** Characteristic of the respondents.

Variable	Category	N	%
**Age**	Mean ± standard deviation	48.5	±15.04
**Gender**	Male	203	38.1
	Female	330	61.9
**Residence**	Village/rural area	125	23.5
	Small town (<20,000 residents)	147	27.6
	Medium-sized city (20,000–100,000 residents)	133	25.0
	Big city (100,000–500,000 residents)	45	8.5
	Biggest city (>500,000 residents)	82	15.4
**Married**	Yes	289	54.2
	No	244	45.8
**Education**	Primary	160	30.0
	Secondary	178	33.4
	University	195	36.6
**Professional situation**	Student	31	5.8
	Working	383	72.0
	Unemployed	31	5.8
	Pensioner	87	16.4
**Self-rated standards of living**	Definitely bad	6	1.1
	Bad	26	4.9
	Rather bad	69	13.0
	No opinion	56	10.5
	Rather good	196	36.8
	Good	137	25.8
	definitely good	42	7.9
**Self-rated health status**	Definitely bad	1	0.2
	Bad	35	6.6
	Rather bad	73	13.7
	No opinion	61	11.5
	Rather good	185	34.8
	Good	142	26.7
	Definitely good	34	6.4
**Interested in healthcare system changes**	Yes	383	72.7
	No	77	14.6
	No opinion	67	12.7
**Frequency of receiving health care**	A few times a month	49	9.2
	Once a month	100	18.8
	Every three months	161	30.3
	Every six months	123	23.2
	Once a year	72	13.6
	Less frequently than once a year	26	4.9
**Form of health care the respondents are most frequently provided with**	Basic health care	369	69.5
	Specialist care in outpatient clinics	107	20.2
	Hospitals	55	10.4

Source: the authors’ own data.

**Table 2 ijerph-18-01178-t002:** Respondents’ opinion on the health care system in Poland, evaluation of the quality and accessibility of health services, factors influencing the effectiveness of the health care system, and factors/areas requiring changes.

Questions	Factors/Areas	Negative Answers N (%)	No Opinion N (%)	Positive Answers N (%)	M	SD
**General assessment of the health care system in Poland**		344 (64.9)	82 (15.5)	104 (19.6)	−0.81	1.31
**To what extent do you agree with the following opinions on the public health care system**	patients are treated with kindness and care	210 (39.7)	44 (8.3)	275 (52)	0.18	1.80
	there are no problems to make an appointment with a primary care doctor	233 (44.1)	44 (8.3)	252 (47.6)	0.03	1.93
	it is easy to obtain information on access to health benefits	238 (45.0)	52 (9.8)	239 (45.2)	−0.06	1.83
	medical treatment is fully free	312 (58.6)	45 (8.5)	175 (32.9)	−0.61	2.00
	treatment conditions are good	236 (44.7)	50 (9.5)	242 (45.8)	−0.06	1.90
	doctors are willing to give referrals to medical specialists if a patient’s condition requires such	176 (33.2)	56 (10.6)	298 (56.2)	0.29	1.80
	patients can expect immediate medical assistance	237 (44.6)	55 (10.4)	239 (45.0)	−0.08	1.91
	all patients are treated equally	275 (52.1)	68 (12.9)	185 (35)	−0.39	1.94
**Evaluation of the quality of health services**	in public institutions of the health care system	258 (48.9)	47 (8.9)	223 (42.2)	−0.22	1.66
	in private institutions of the health care system	48 (9.1)	65 (12.3)	415 (78.6)	1.30	1.36
**Evaluation of the accessibility of health services**	in public institutions of the health care system	310 (58.8)	48 (9.1)	169 (32.1)	−0.68	1.67
	in private institutions of the health care system	44 (8.3)	62 (11.7)	425 (80.0)	1.32	1.31
**Factors influencing the effectiveness of the health care system**	organization of the health care system	77 (14.5)	52 (9.8)	403 (75.8)	1.32	1.68
	financing	55 (10.3)	53 (10)	424 (79.7)	1.59	1.64
	number of practicing doctors	74 (13.9)	63 (11.8)	396 (74.3)	1.31	1.73
	competences of practicing doctors	82 (15.5)	51 (9.6)	396 (74.9)	1.25	1.78
	hospital infrastructure	78 (14.7)	68 (12.8)	386 (72.6)	1.20	1.67
	medical equipment in diagnostics and therapy	62 (11.7)	63 (11.9)	405 (764)	1.38	1.66
	costs of medications	86 (16.1)	66 (12.4)	381 (71.5)	1.15	1.84
	prevention/health education	73 (13.8)	51 (9.6)	405 (76.6)	1.31	1.65
**Factors/areas requiring changes**	system of training of medical personnel	78 (14.8)	76 (14.4)	374 (70.8)	1.09	1.46
	financing of health services	37 (7.0)	38 (7.2)	452 (85.8)	1.69	1.32
	limited role of the state in the decision-making process in the system	51 (9.6)	102 (19.3)	376 (71.1)	1.28	1.47
	role of health insurance	25 (4.7)	79 (15.0)	423 (80.3)	1.39	1.15
	free-market rules	48 (9.1)	140 (26.5)	341 (64.5)	0.90	1.26
	priorities in the health care system	29 (5.5)	75 (14.2)	423 (80.3)	1.56	1.28

Source: the authors’ own data.

## Data Availability

The data presented in this study are available on request from the corresponding author.

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
