# Peer review of "Priority Setting in the Polish Health Care System According to Patients’ Perspective"

_ijerph, 2021, doi:10.3390/ijerph18031178_

Round 1
Reviewer 1 Report
Thank you for the opportunity to review this manuscript. The topic of the article is important for the readers of IJERPH.
Authors tried to present the patients perspectives of the priority setting in the Polish health care system, but:
- In the Introduction part there is no information regarding the main characteristicts, current problems or challenges of the Polish health care system. There is also a lack of information regarding patients associantions/organisation and their activities and role in the Polish health care system.
- Methods:
- the study was conducted among patienst. Was the research methodology approved by the Bioethical Committee?
- when and who conducted the study?
- information regaarding the study tool is too general and no reference was given.
- Results: Auhors have wtitten: Considering the wide scope of the questions included in the research tool, the article discusses
selected results only (lines 81-82). The list of questions is not presented, so the choice is not clear. In my opinion the presentation of the results is very general. There is lack of identifications of any relationship among analysed items. - Discussion. The results obtained in the study are not properly discussed in this part of the manuscript. The number of references reagrding the topic is also scarce.
Author Response
Dear Reviewer,
Thank you for taking the time to review our manuscript and all comments! They certainly allowed us to improve our paper.
All comments have been deeply analyzed and entered into the manuscript. Please see the attachment.
In this place, point-by-point we would like to answer for your comments and suggestions:
- In the Introduction part there is no information regarding the main characteristicts, current problems or challenges of the Polish health care system. There is also a lack of information regarding patients associantions/organisation and their activities and role in the Polish health care system.
We introduced significant changes in the “introductory part”, added the Polish context, added some problems and challenges of the Polish health care system.
“Reforming the health care system is a complex social, political and economic process based on set aims and attempts made to achieve them so that greater attention to the practice and consequences of priority setting are required to promote accountability, efficiency, effectiveness and equity [1,2]. Priority setting determines the sustainability of any health system, whether primarily publicly or privately financed, and so is one of the greatest challenges faced by policy makers [3,4]. The term priority often come up in discussions on health issues. A health priority is interpreted as an instruction to help define the most critical activities, at a given time and in specific circumstances. Adopting one or more priorities signals the importance of the tasks required to accomplish the priority and helps direct the work of an organization or system [5]. Prioritising takes place in all parts of the health care system where demands and needs exceed resources. Decisions on priority setting are made at different levels [4-7]. General policy decisions are made at national and regional levels, as are comprehensive decisions on resource allocation, systems for financing providers, and national guidelines including priority setting for management of common diseases [8]. It guides investments in health care and health research, and respects resource constraints. It happens continuously, with or without appropriate tools or processes. The use of scientific evidence and principles in setting health priorities has an enormous potential to lead to more rational decision making, especially in low resource settings where decision making has long lacked formal tools, processes, or an evidence base [9-11].
Health policy in Poland is changed, the quality of services is increased, and also the level of financing, mainly from public benefits. Despite constant growth of indexes reflecting the health status of Polish society, such as life expectancy, quality of life, or decreasing index of deaths at birth, just as in the majority of European countries, in Poland the society is growing older, which implies the necessity to reorganize the system. Polish health care is far from being ideal. In the rankings in comparison with other countries, also the neighboring ones, its picture is bad. Patients, physicians themselves, and politicians complain about it [12]. The present situation of the Polish health care system has been and still is influenced by numerous political, market, economic, and human factors. The changes that have been introduced still do not seem to fully include the superior role of the patient in the system. Presently, the most important influence on the health care system seems to be economic and demographic factors.[13]. Despite this, the primary objective of the health care system in Poland is to guarantee all citizens an equal access to health services in accordance with the principles of social solidarity and equity. Polish health care priorities are established mainly to insure this equitable distribution of resources, which is an important part of strategic planning [14]. However, a properly functioning health care system must also prioritize aspects of health care delivery valued by patients, as equity of access and resource distribution are achieved. With social solidarity and equity being overarching priorities of the nation, the health care system must define its own priorities, with larger national goals, that will deliver quality care and meet the expectations of consumers [13]. Because defined priorities drive organizational performance, they must be carefully selected. The question has been asked who should be involved in this process of defining priorities and what method will produce the most effective and ethical results. Involving all the stakeholders in the system, including the public and local community, in selecting priorities especially in publicly funded programs and initiatives [15], is an ideal solution. Including input from all the stakeholders of the system, could have a significant impact on the quality and efficiency of the health care system”.
- Methods:
- the study was conducted among patienst. Was the research methodology approved by the Bioethical Committee?
- when and who conducted the study?
- information regaarding the study tool is too general and no reference was given.
We added:
“The pilot study on 22 people was carried out in May 2017. Next, a face-to-face interview questionnaire was conducted with 533 respondents (18 years of age and older) who agreed to participate in this research, on September 2017 - March 2018, in four public health care units in the Lodz region. The respondents were a hospital patients, stay there longer than 3 days and had a conditions adequate to take part in this research (aware, no pain, comfortably). The participation was voluntary and anonymous. All the subjects gave their informed consent before they participated in the study. The interviews took from 20 to 30 minutes.
The interviewers had the directors’ and head’ of wards written consent to conduct the research in hospitals.
The research tool was a questionnaire prepared by the staff of the Department of Health Care Policy and the Centre for Research on Health Care Strategies and Health Policy at the Warsaw School of Economics in cooperation with the National Medical University in Kiev.
The questionnaire consists of twenty four questions: six concerned the demographic characteristic of respondents, i.e the independent variables such as age, gender, residence, civil status, education, professional situation. Next six questions were connected with self-reported standards of living and health assessment, interested in healthcare system changes, frequency of receiving health care and form of health care the respondents are most frequently provided with. Questions about patients’ opinion of optimal solutions for organization, functioning and financing of the health care system in Poland were the second, more expandable part of the questionnaire.
Each of the listed categories was assessed on a 7-pont Likert scale from “definitely bad/ definitely no/” (-3 points), “no” (-2 points), “rather no” (-1 point), “no opinion” (0), “rather yes” (1 point), “yes” (2 points), “definitely good/ definitely yes” (3 point). There were no open questions in the questionnaire. The questionnaire was evaluated in terms of compliance with the main rules for questionnaire construction, i.e.,: simple vocabulary, avoiding words that are abstract, not fully defined or ambiguous, jargon, moralizing language, avoiding negations, names of institutions, surnames, too long items”.
- Results: Auhors have wtitten: Considering the wide scope of the questions included in the research tool, the article discusses selected results only (lines 81-82). The list of questions is not presented, so the choice is not clear. In my opinion the presentation of the results is very general. There is lack of identifications of any relationship among analysed items.
In “Materials and Methods section” we added this information. We hope that there are sufficient.
- The results obtained in the study are not properly discussed in this part of the manuscript. The number of references reagrding the topic is also scarce.
The discussion section has been shortened, we changed references and focused only on the key points.
We added:
“The priority setting in health care remains a crucial part of health care planning and resources allocation but also it is a point the patients pay their attention to in practice from the perspective of own experiences and observations [15-17]. The approach adopted in the article differs from the patients satisfaction survey in that it focuses on “everyday” beneficiary patients reflections coming from their experiences in using health services concerning care, organization of service delivery, financing, accessibility to services, quality of services. Moreover, patients' opinions on potential changes in the health care system in Poland allow for the disclosure of their active attitudes towards the current state of the system. Although the obtained results are not surprising for some researchers and health care stakeholders they have a value as the evidence of the patients opinions in terms of time, and as the starting point of the process aiming at meeting public expectations and setting directions of further research. However, it is difficult to separate these areas dichotomously.
The obtained results of the survey of patients' opinions may make it possible to identify effective, friendly for patients solutions in health care, but also allow to recognize limitations that constitute obstacles to the use of health services [18]. Considering the opinions in which over half of the respondents negatively evaluated the Polish health care system as a whole and a great majority of the respondents express that aspects under study require immediate improvement proves dissatisfaction with the way the health care is delivered especially in terms of payments for health care and treating patients equally. Patients opinions on the more detailed issues of the health care system indicating the key problems from their own point of view may facilitate the interpretation of the overall opinion on the health care system. The gradual refinement of questions included in the questionnaire may be helpful in better understanding the respondents views and enables interpretation of the reasons of their answers [19].
The tasks of the state authorities aimed at providing equal access to health in Poland are specified in the Law of 27 August 2004 on Health Care Benefits Financed from Public Funds [20]. These tasks include providing conditions for functioning of the health care system; analysis and identification of health needs and factors changing them; health promotion and prevention focused on creating conditions favourable for health; financing. It needs to be emphasized that an element of the health care system which has an impact on its functioning is financing [21, 22]. The most individuals express the opinion that financing is a factor which has the greatest influence on the effectiveness of the system and believe that the rules of financing of health benefits require radical changes. Consequently, they share the opinion that the role of health insurance should be changed radically. Given the patients perception and results of health care accessibility research [23], it could be stated that the amount of money spending on health care in Poland does not translate into accessibility of health benefits or subjective assessment of satisfying medical needs of Polish patients. Health care stakeholders debate on the subject of resources allocation can be reinitiated.
Patients also pay particular attention to medical equipment used in diagnostics and therapy and prevention and education as the important factors influencing effectiveness of health care. While the first factor seems to be related to the accessibility to health technologies, which in turn may have various reasons (financial, organizational, administrative) the second factor expresses quite different value for patients to improve their knowledge in the field of health protection. It is reflected in the active attitude of the respondents towards factors and areas requiring changes, mainly training medical personnel, financing, insurance, priorities.
The opinions of the society members can help to identify health care areas for action of policy maker, medical professionals, and other bodies responsible for setting health care priorities and can contribute to the establishment directions of redefining methods and instruments contributing to increasing patient satisfaction. However, we should bear in mind the fundamental difference between patient satisfaction and meeting their expectations in terms of their health needs [24-30].
Moreover, the feedback obtained from patients may inspire to create methods of promoting, convincing patients to the nationally chosen priorities by dialogue platform or disseminating knowledge on opportunities and limitations the health care system may meet assuming scarce system resources [31]. It seems to be indisputable that the assumptions of the future health policy should be formulated in the context of results of the survey that reflects opinions of different social groups, including also or even most of all, beneficiaries of the health care system”.
We hope that you will find our manuscript acceptable in its present form, but if you have other remarks, please contact us. Again, thank you very much for your attention.
Sincerely yours,
Authors
Reviewer 2 Report
This manuscript presents findings from a survey of hospital patients regarding the state of the Polish health care system.
The survey methodology is poorly explained, so the reader has no information on how many patients were invited to participate in the survey, what percent completed a survey, and how representative the survey participants were of the region from which they were sampled.
Survey participants selected responses along a seven-point scale. However, these variables were treated as categorical. Authors should provide a mean and standard deviation for these scale variables, not percentages according to seven categories.
The manuscript needs only one table to describe survey participants and one table to describe survey results (not one table per survey question).
The authors use too many words to get to their point and the meaning is lost in all the unnecessary words. The discussion section is especially cumbersome. It should be dramatically shortened, and should focus on the key points.
The limitations section likely needs to include limitations regarding how the sample was selected (e.g., was it a convenience sample) and how representative it is. Small numbers are one issue, but there are many other concerns with the methodology.
I have uploaded a pdf with some suggested edits to improve the English and better convey meaning. Suggestions are also provided in comments.

Author Response
Dear Reviewer,
Thank you for taking the time to review our manuscript and all comments! They certainly allowed us to improve our paper. It’s very kind of you to uploaded a pdf with some suggested edits to improve the English and better convey meaning.
All comments have been deeply analyzed and entered into the manuscript. Please see the attachment.
In this place, point by point we would like to answer for your comments and suggestions:
- The survey methodology is poorly explained, so the reader has no information on how many patients were invited to participate in the survey, what percent completed a survey, and how representative the survey participants were of the region from which they were sampled.
We added:
“The pilot study on 22 people was carried out in May 2017. Next, a face-to-face interview questionnaire was conducted with 533 respondents (18 years of age and older) who agreed to participate in this research, on September 2017 - March 2018, in four public health care units in the Lodz region. The respondents were a hospital patients, stay there longer than 3 days and had a conditions adequate to take part in this research (aware, no pain, comfortably). The participation was voluntary and anonymous. All the subjects gave their informed consent before they participated in the study. The interviews took from 20 to 30 minutes.
The interviewers had the directors’ and head’ of wards written consent to conduct the research in hospitals.
The research tool was a questionnaire prepared by the staff of the Department of Health Care Policy and the Centre for Research on Health Care Strategies and Health Policy at the Warsaw School of Economics in cooperation with the National Medical University in Kiev.
The questionnaire consists of twenty four questions: six concerned the demographic characteristic of respondents, i.e the independent variables such as age, gender, residence, civil status, education, professional situation. Next six questions were connected with self-reported standards of living and health assessment, interested in healthcare system changes, frequency of receiving health care and form of health care the respondents are most frequently provided with. Questions about patients’ opinion of optimal solutions for organization, functioning and financing of the health care system in Poland were the second, more expandable part of the questionnaire.
Each of the listed categories was assessed on a 7-pont Likert scale from “definitely bad/ definitely no/” (-3 points), “no” (-2 points), “rather no” (-1 point), “no opinion” (0), “rather yes” (1 point), “yes” (2 points), “definitely good/ definitely yes” (3 point). There were no open questions in the questionnaire. The questionnaire was evaluated in terms of compliance with the main rules for questionnaire construction, i.e.,: simple vocabulary, avoiding words that are abstract, not fully defined or ambiguous, jargon, moralizing language, avoiding negations, names of institutions, surnames, too long items”.
- Survey participants selected responses along a seven-point scale. However, these variables were treated as categorical. Authors should provide a mean and standard deviation for these scale variables, not percentages according to seven categories.
Thank you for this suggestion. Mean and standard deviation were calculated and presented in table 2.
- The manuscript needs only one table to describe survey participants and one table to describe survey results (not one table per survey question).
We added one table to describe survey participants and one table to describe survey results. I was very good suggestions. We think that the presentation of our results is much clearer.
- The authors use too many words to get to their point and the meaning is lost in all the unnecessary words. The discussion section is especially cumbersome. It should be dramatically shortened, and should focus on the key points.
The discussion section has been shortened, we changed references and focused only on the key points.
We added:
“The priority setting in health care remains a crucial part of health care planning and resources allocation but also it is a point the patients pay their attention to in practice from the perspective of own experiences and observations [15-17]. The approach adopted in the article differs from the patients satisfaction survey in that it focuses on “everyday” beneficiary patients reflections coming from their experiences in using health services concerning care, organization of service delivery, financing, accessibility to services, quality of services. Moreover, patients' opinions on potential changes in the health care system in Poland allow for the disclosure of their active attitudes towards the current state of the system. Although the obtained results are not surprising for some researchers and health care stakeholders they have a value as the evidence of the patients opinions in terms of time, and as the starting point of the process aiming at meeting public expectations and setting directions of further research. However, it is difficult to separate these areas dichotomously.
The obtained results of the survey of patients' opinions may make it possible to identify effective, friendly for patients solutions in health care, but also allow to recognize limitations that constitute obstacles to the use of health services [18]. Considering the opinions in which over half of the respondents negatively evaluated the Polish health care system as a whole and a great majority of the respondents express that aspects under study require immediate improvement proves dissatisfaction with the way the health care is delivered especially in terms of payments for health care and treating patients equally. Patients opinions on the more detailed issues of the health care system indicating the key problems from their own point of view may facilitate the interpretation of the overall opinion on the health care system. The gradual refinement of questions included in the questionnaire may be helpful in better understanding the respondents views and enables interpretation of the reasons of their answers [19].
The tasks of the state authorities aimed at providing equal access to health in Poland are specified in the Law of 27 August 2004 on Health Care Benefits Financed from Public Funds [20]. These tasks include providing conditions for functioning of the health care system; analysis and identification of health needs and factors changing them; health promotion and prevention focused on creating conditions favourable for health; financing. It needs to be emphasized that an element of the health care system which has an impact on its functioning is financing [21, 22]. The most individuals express the opinion that financing is a factor which has the greatest influence on the effectiveness of the system and believe that the rules of financing of health benefits require radical changes. Consequently, they share the opinion that the role of health insurance should be changed radically. Given the patients perception and results of health care accessibility research [23], it could be stated that the amount of money spending on health care in Poland does not translate into accessibility of health benefits or subjective assessment of satisfying medical needs of Polish patients. Health care stakeholders debate on the subject of resources allocation can be reinitiated.
Patients also pay particular attention to medical equipment used in diagnostics and therapy and prevention and education as the important factors influencing effectiveness of health care. While the first factor seems to be related to the accessibility to health technologies, which in turn may have various reasons (financial, organizational, administrative) the second factor expresses quite different value for patients to improve their knowledge in the field of health protection. It is reflected in the active attitude of the respondents towards factors and areas requiring changes, mainly training medical personnel, financing, insurance, priorities.
The opinions of the society members can help to identify health care areas for action of policy maker, medical professionals, and other bodies responsible for setting health care priorities and can contribute to the establishment directions of redefining methods and instruments contributing to increasing patient satisfaction. However, we should bear in mind the fundamental difference between patient satisfaction and meeting their expectations in terms of their health needs [24-30].
Moreover, the feedback obtained from patients may inspire to create methods of promoting, convincing patients to the nationally chosen priorities by dialogue platform or disseminating knowledge on opportunities and limitations the health care system may meet assuming scarce system resources [31]. It seems to be indisputable that the assumptions of the future health policy should be formulated in the context of results of the survey that reflects opinions of different social groups, including also or even most of all, beneficiaries of the health care system”.
- The limitations section likely needs to include limitations regarding how the sample was selected (e.g., was it a convenience sample) and how representative it is. Small numbers are one issue, but there are many other concerns with the methodology.
We added:
“Our study had limitations. First, we have described priority setting as perceived by patients, which may not reflect the objective priority-setting process. However, the information from our participants may be an indicator of the degree of publicity of the priority-setting process, which in itself is a relevant finding to this study. Second, we are not able to generalize our findings but they provided us with insight in the priority-setting of the cases we described. Third, the measure of knowledge maybe imprecise due to the small number of questions in the survey. This issue requires further research”.
We hope that you will find our manuscript acceptable in its present form but if you have other remarks, please contact us. Again, thank you very much for your attention.
Sincerely yours,
Authors
Round 2
Reviewer 1 Report
Thank you very much for the revision of the manuscript.
In my opinion the study tools (the questionnaire and interview scenario) could be attached as Supplementary materials? Please consider this option.
Author Response
Dear Reviewer,
Again, thank you very much for your comments and attention.
When we were preparing the manuscript, we considered adding the questionnaire as an appendix, but it was not added in the final version. After your suggestion, the questionnaire has been translated and we will attach it to the article. However, we think that the editor should make a final decision.
Thank you very much for your valuable suggestion.
We hope that you will find our manuscript acceptable in its present form.
Sincerely yours,
Authors

Reviewer 2 Report
The authors have done a nice job revising this paper and presenting the results.
Before the reader can assess the value of this set of survey respondents, it is critical to know the universe from which the sample was selected, the response rate, and how the sample compares to the universe and to the region.
For example, you should include something like the following: "All persons with a hospital stay lasting at least 3-days between January and June 2019 were invited to participate in the survey. Of the 1500 persons who were invited to participate, 500 (33%) agreed to participate and completed a survey. Participants were 65% female and 35% male, compared to 45% female and 55% male for the hospitalized population, and 52% female and 48% male for the regional population."
Additionally, there are far too many words in this paper to effectively convey meaning. Consider this original paragraph and the revised, much shorter and clearer paragraph (which uses less than half the number of words without losing any information):
Original paragraph: “Reforming the health care system is a complex social, political and economic process based on set aims and attempts made to achieve them so that greater attention to the practice and consequences of priority setting are required to promote accountability, efficiency, effectiveness and equity [1,2]. Priority setting determines the sustainability of any health system, whether primarily publicly or privately financed, and so is one of the greatest challenges faced by policy makers [3,4]. The term priority often come up in discussions on health issues. A health priority is interpreted as an instruction to help define the most critical activities, at a given time and in specific circumstances. Adopting one or more priorities signals the importance of the tasks required to accomplish the priority and helps direct the work of an organization or system [5]. Prioritising takes place in all parts of the health care system where demands and needs exceed resources. Decisions on priority setting are made at different levels [4-7]. General policy decisions are made at national and regional levels, as are comprehensive decisions on resource allocation, systems for financing providers, and national guidelines including priority setting for management of common diseases [8]. It guides investments in health care and health research, and respects resource constraints. It happens continuously, with or without appropriate tools or processes. The use of scientific evidence and principles in setting health priorities has an enormous potential to lead to more rational decision making, especially in low resource settings where decision making has long lacked formal tools, processes, or an evidence base [9-11].”
Revised, shortened paragraph: “Reforming the health care system involves social, political and economic considerations. It requires a robust process to efficiently and effectively establish priorities to guide the reform, and to ensure accountability and equity [1,2]. Establishing priorities is one of the greatest challenges faced by policy makers [3,4]. A priority is an instruction to help define the most critical activities, and signals the important work of an organization or system [5]. Prioritising is essential when demands and needs exceed resources. Priority setting decisions are made at different levels [4-7], including national, regional, system and institution [8], and often lack appropriate tools, information and processes. The use of scientific evidence and principles in setting health priorities has enormous potential to lead to more rational decision making, especially in low resource settings [9-11].”
In general, the paper can be dramatically tightened (and shortened). For example, we don't need to know how many questions were contained in each section of the questionnaire, and not all the results need to be presented. Pull out the key results only, that speak directly to priority setting (in addition to the demographics).
Author Response
Dear Reviewer,
Again, thank you very much for your valuable comments and suggestions.
All comments have been deeply analyzed.
In this place, point by point we would like to answer for your comments and suggestions:
- Before the reader can assess the value of this set of survey respondents, it is critical to know the universe from which the sample was selected, the response rate, and how the sample compares to the universe and to the region. For example, you should include something like the following: "All persons with a hospital stay lasting at least 3-days between January and June 2019 were invited to participate in the survey. Of the 1500 persons who were invited to participate, 500 (33%) agreed to participate and completed a survey. Participants were 65% female and 35% male, compared to 45% female and 55% male for the hospitalized population, and 52% female and 48% male for the regional population."
Several reasons decided to conduct the study in a selected group of patients. The study was the research stage aimed at improving the research tool and obtaining the highest possible number of answers within the existing financial constraints. Before the further research on a randomly selected sample (the research was not financed from external funds). The aim of the study was also to learn about patients' opinions dominant in the structure of responses in order to possibly strengthen the role of some of the questions asked in the future questionnaire. We selected 4 hospitals from the Lodz region, the selection of which was also important due to the favorable attitude of their management. A goal of the study was to include as many participants as possible in order to 500 . At the initial stage, no formal calculation of sample size was carried out but we choose group of more than 100 patients in each of the hospitals. Similar number of patients was in every group. The patients were selected according to the accepted criteria: they did not feel pain, they felt relatively well and agreed to participate in our study. The research protocol did not indicate how many patients refused to participate in the study. The study was conducted from September 2017 to March 2018.
If you think this are important information of this study we can add this in manuscript.
- Additionally, there are far too many words in this paper to effectively convey meaning. Consider this original paragraph and the revised, much shorter and clearer paragraph (which uses less than half the number of words without losing any information):
The manuscript was revised by the Translation Center of the Medical University of Lodz and slightly shortened. Please see the attachment
- ”In general, the paper can be dramatically tightened (and shortened). For example, we don't need to know how many questions were contained in each section of the questionnaire, and not all the results need to be presented. Pull out the key results only, that speak directly to priority setting (in addition to the demographics).
The other reviewer stated in his review that the information in the first version of the manuscript on the research tools are too general. For this reason, the questionnaire has been described in detail. Additionally, we were suggested to add the questionnaire as an attachment. It has been translated and added to the current version.
Only some part of the results was presented in the manuscript, according to us those that relate most to priorities.
We hope that you will find our manuscript acceptable in its present form.
Again, thank you very much for your attention.
Sincerely yours,
Authors